# Physical Activity versus Selected Health Behaviors, Subjective Physical and Psychological Health and Multimorbidity in a Large Cohort of Polish Seniors during the COVID-19 Pandemic (Results of the National Test for Poles’ Health )

**DOI:** 10.3390/ijerph20010556

**Published:** 2022-12-29

**Authors:** Agnieszka Szychowska, Anna Zimny-Zając, Elżbieta Dziankowska-Zaborszczyk, Tomasz Grodzicki, Wojciech Drygas, Tomasz Zdrojewski

**Affiliations:** 1Department of Social and Preventive Medicine, Medical University of Łódź, Żeligowskiego St. 7/9, 90-752 Łódź, Poland; 2Medonet, Ringier Axel Springer Poland, Domaniewska St. 49, 02-672 Warsaw, Poland; 3Department of Epidemiology and Biostatistics, Social and Preventive Medicine, Medical University of Łódź, Żeligowskiego St. 7/9, 90-752 Łódź, Poland; 4Department of Internal Medicine and Gerontology, Jagiellonian University Medical College, 2 Jakubowskiego St., 31-501 Kraków, Poland; 5Department of Epidemiology, Cardiovascular Disease Prevention and Health Promotion, National Institute of Cardiology, Alpejska St. 42, 04-628 Warsaw, Poland; 6Department of Preventive Medicine and Education, Medical University of Gdańsk, Dębinki 7, 80-211 Gdańsk, Poland

**Keywords:** healthy aging, aging, physical activity, exercise, lifestyle, health behavior

## Abstract

National Test for Poles’ Health is an online study conducted on a large group of Polish Internet users. For the purpose of this study, 64,732 subjects (48.8% female) over 65 years old were included. Subjects provided answers on the level of physical activity (PA) they engage in, prevalence of non-communicable diseases (obesity, hypertension, diabetes, heart diseases, chronic obstructive pulmonary disease (COPD), depression, cancer) and subjective physical and psychological health. Additionally, their Body Mass Index (BMI) and prevalence of multimorbidity was assessed. We found that older people who engage in at least 2 h of physical activity/week had significantly lower prevalence of hypertension, obesity and heart diseases than those who engaged in 1–1.5 h/week or less than 1 h/week. Multimorbidity was present in 33.2% of subjects from the most active group and 52.6% of the least active ones. Subjective physical and psychological health was rated as “very good” by 26.6% and 41.2%, respectively, by subjects from the most active group. Only 9.1% of the least active subjects rated their physical health as “very good” and only 27.4% rated their psychological health as such. Regular physical activity may be a helpful tool in combating the reduced well-being of older people affected by the isolation caused by the COVID-19 pandemic. Unfortunately, over 65% of respondents claimed to engage in less than 1 h of PA a week or less.

## 1. Introduction

No population before has had the life expectancy as high as people alive today. The global population of older people has grown more than 5.5 times over the past 70 years, and it is predicted to reach nearly 1.5 billion by 2050 (16% of the expected global population) [1]. Therefore, it is constantly becoming increasingly important to focus on the health and well-being of this large part of the population. Projections from Polish Central Statistical Office (GUS) show an expected increase in the number of older people living in Poland in the next years. Percentage of people aged 65 years and over will increase from 18.6% in 2020 to 24.5% in 2035 [2,3]. Similar to many other European countries, Poland faces the challenge of a growing population of older people, who have a higher risk of developing chronic diseases, often lifestyle-related. The GUS studies have also shown that the majority of older people in Poland do not engage in sufficient physical activity [4], which is also a problem in other countries, such as United Kingdom [5] or the United States [6]. Regular exercise has been shown to have many benefits for physical and mental health, as well as self-rated quality of life of older people. Engaging in physical activity lowers the risk of cardiovascular disease (CVD) [7], diabetes [8], hypertension [9], frailty [10], dementia [11] and other chronic diseases often associated with older age. It is one of the most crucial factors in active and healthy aging [12,13]. Healthy aging means good quality of life, independence, freedom from disability and high social engagement of older people; therefore, it should be a priority for public health policymakers.

As we are in the third year of a global pandemic, it is evident that older people are among those most affected by it. Older people are in the higher risk group for developing severe symptoms as well as death from COVID-19 [14]. Accompanying diseases, such as hypertension, diabetes, CVD which increase the risk of severe symptoms and death from COVID-19 are caused, in part, by lifestyle choices, such as lack of physical activity. Regular exercise has been shown to help strengthen one’s immune system by modulating the release of pro- and anti-inflammatory cytokines, increasing lymphocyte circulation and cell recruitment, which is associated with lower risk of mortality or developing severe symptoms in viral infections [15]. However, maintaining a healthy lifestyle during the pandemic has shown to be more difficult, due to quarantine and other necessary restrictions [16].

National Test for Poles’ Health (Narodowy Test Zdrowia Polaków, NTZP) is a study conducted by Medonet (an immensely popular health-oriented Internet platform) on a large group of Polish Internet users. It includes an online questionnaire focused on a variety of health behaviors among the respondents (for example, diet, exercise, sleep, frequency of medical consultations, smoking, use of alcohol and drugs and other). NTZP questionnaire is constructed to be understandable for everyone and simple to fill out. After subjecting their answers, respondents are presented with a score, a health index, indicating their level of health and some recommendations on how to improve it. They also obtain an estimation of their biological age, which serves as a clue on how to improve their lifestyle and possibly slow down their biological aging; it is based on an algorithm by Thomas Perls from Boston University School of Medicine [17,18]. NTZP is a valuable source of information on the health of Polish Internet users and, so far, it has been conducted in three waves (2020, 2021 and 2022).

The aim of this article is to present some of the findings of one of the biggest studies on the health status of Polish seniors. This article focuses on selected health behaviors, as well as a subjective assessment of one’s health and the prevalence of multimorbidity in relation to the level of physical activity.

## 2. Materials and Methods

The questionnaire was filled out by over 300,000 adult respondents in both waves of the NTZP. The second wave was conducted a year after the first pandemic-related restrictions were implemented in Poland. At that time (end of March 2021), there were 187,526 confirmed cases and 3055 deaths weekly from COVID-19 in Poland (according to the WHO database). For the purpose of this article, responses from the second wave (2021) were analyzed. Further, for the purpose of this analysis, only responses from people aged 65 years and over (65 to 99 years) were included (over 64,000 subjects). Mean age was 70.1 and median 69.0 years old (standard deviation = 4.5). Further characteristics of the subjects included in this study can be found in Table 1.

Respondents were divided into three groups depending on the amount of physical activity they engaged in: high level of physical activity (at least 2 h/week), medium (1–1.5 h/week). The third group included people who do not exercise at all and people who engage in less than 1 h of physical activity a week, since these two subgroups showed very similar results. Physical activity was defined as engaging in any type of sport; a separate question was asked about physical activity in the form of walking. For the purpose of this analysis, physical activity in the form of active sport participation was included. Body Mass Index (BMI) was calculated based on the respondents’ answers about their weight and height.

Respondents were asked whether they have ever been diagnosed with following non-communicable diseases: obesity, hypertension, diabetes, heart diseases, chronic obstructive pulmonary disease (COPD), depression, cancer.

Respondents rated their subjective physical and psychological health (compared to other people their age) on a scale from “very good” to “very bad”, with five possible answers to choose from.

The data from NTZP are available to the public in the form of reports published on the website narodowytestzdrowia.medonet.pl.

The authors have provided Appendix A in the form of the questionnaire and database.

### Statistical Analysis

The subjects’ characteristics were assessed using descriptive statistics. For measurable features, their minimum and maximum values, arithmetic mean values with corresponding 95% confidence intervals and values of standard deviations were calculated. In the case of qualitative variables, the percentages of their respective categories and the corresponding 95% confidence intervals were calculated. Taking into account the fact that for large groups of several hundred units, it can be concluded that the distribution of features is close to the normal distribution; to compare the significance of differences in mean values in two groups, the Student’s *t*-test for independent samples was used, and in three groups, the analysis of variance ANOVA with the Sheffe post-hoc test was used. To assess the significance of differences between the frequencies of the categories of non-measurable variables in two groups, the test for two fractions from large samples was used. The differences were considered statistically significant for *p* < 0.05. All analyses were performed using the statistical software package Statistica, version 13.1, manufactured by Dell Inc. (Tulsa, OK, USA).

## 3. Results

More than a quarter (25.5%) of men aged 65 and older exercised for at least 2 h/week. In women, the prevalence of high level of physical activity was lower: 18.1% The differences were statistically significant (*p* < 0.001). Similarly, more men than women exercised for 1–1.5 h/week (13.3% vs. 12.7%, *p* < 0.05). A total of 69.2% of women exercised for less than 1 h/week or did not exercise regularly. The same situation was present in 61.2% of men. This difference was also statistically significant with *p* < 0.001. These findings are presented in Table 2.

Physical activity (active sport participation) of the respondents was related to the time they spent daily walking (as a form of recreation and rest or as a form of reaching a certain destination, work, home, etc.). Physically active people more often walked longer, while less active respondents more often walked shorter or did not leave the house at all (Χ^2^ = 4935.385, *p* = 0.0000). Further information can be found in Table 3.

All groups had a mean BMI > 25 kg/m^2^, as can be seen in Table 4, Table 5 and Table 6. However, mean BMI was lower the higher the level of physical activity. The difference between mean BMI in the most active and the least active group was 1.46 kg/m^2^ and the difference was statistically significant (*p* < 0.001). Comparing the differences between women and men, we can observe that the mean BMI was lower in women in all three groups of physical activity (high activity: t = 12.023, *p* = 0.000000; medium activity: t = 9.001, *p* = 0.000000; low activity: t = 12.869, *p* = 0.000000; all: t = 15.555, *p* = 0.000000).

In people aged 65 years and over, hypertension was the most prevalent non-communicable disease (57.1% of respondents). The second most frequently diagnosed was obesity (28.1% of respondents), followed by heart diseases (almost 26%). COPD (6.3%), depression (7.8%) and cancer (10%) were less prevalent. A similar situation was observed in all groups of physical activity with one exception: people with the highest level of physical activity were less likely to have obesity (18.2%) than heart diseases (20%). Table 7 presents this data.

Multimorbidity, defined as the presence of two or more non-communicable diseases (from the seven non-communicable diseases listed above), was prevalent in 46.7% of all respondents aged 65 years and older. However, it was the least prevalent in the most active group (33,2%, *p* < 0.001). Women were less likely to have multimorbidity when compared to men in all groups of physical activity, as can be seen in Table 8.

In the most active and medium active group, the biggest percentage of respondents was diagnosed with one disease (out of seven mentioned) (35.9% and 35.0%, respectively, the difference was not statistically significant), followed by no diseases diagnosed (30.9% and 24.9%, *p* < 0.001). In the group that did not engage in significant levels of physical activity, the most often answer was one disease diagnosed (29.8%), but the second most often answer was two diagnosed diseases (27.7%). Only 17.6% of subjects in this group had no diagnosis of the mentioned diseases. These findings are presented in Figure 1.

Half (50.7%) of respondents from the most active group rated their physical health as “good” and 26.6% as “very good”. Similarly, in the medium physical activity group, half of the respondents (50.7%) rated their physical health as “good”, followed by 31.3% who rated it as “average (neither good, nor bad)”. Only 15.2% from this group thought their physical health was very good. The biggest part of the least active group (42.9%) rated their physical health as “average”, 39.8% as “good” and only 9.1% as “very good”. Figure 2 presents these findings.

Subjective psychological health was rated as “very good” by 41.2% of respondents from the most active group, 33.1% from the medium group and 27.4% from the least active group (*p* < 0.001), as can be observed in Figure 3. “Average” psychological health was reported by 20.8% of respondents from the least active group but only 11.8% from the most active group (*p* < 0.001).

## 4. Discussion

In this study, it can be observed that people over 65 years old who engaged in some level of physical activity had significantly better rates of subjective physical and psychological health, less prevalent multimorbidity and lower BMI. These active groups also showed smaller prevalence of all seven non-communicable diseases when compared to the group that engaged in less than an hour of exercise a week. Therefore, this study’s findings support previous studies that suggest physical activity can have a protective effect on the health of older people [18].

More active respondents had a mean BMI lower than the least active ones. It has been acknowledged that low body weight (being underweight) has been shown to be more associated with higher risk of mortality in older people [19]. However, a study using AI (artificial intelligence) to assess the risk of mortality in older adults showed that a high normal weight is the most optimal state and not being overweight [20]. It has been shown that BMI indicating obesity in older adults can be seen as a protective factor against osteoporosis and injuries from falls [21], which has been described as the “obesity paradox”. However, this belief that obesity has favorable effect on bone density has also been challenged [22].

Multimorbidity passes a great challenge in treating older patients and can be often the cause for polypharmacy [23]. A higher risk of multimorbidity is associated with older age, female sex, lower socio-economical status and education level, smoking, obesity and lower levels of physical activity [24]. Findings from this study showed that, indeed, less physically active subjects tended to have a higher prevalence of multimorbidity and higher number of diseases diagnosed. Even though female sex has been identified as a risk factor for multimorbidity [25], in this study women had lower percentages of multimorbidity prevalence in all three groups when compared to men. Another study focusing on the health of older people in Poland—PolSenior2, has found a prevalence of multimorbidity of 69.3% in the youngest age group (60–64 years old) up to 90% in people aged 80–89 years [26]. In this study, multimorbidity has been found to be present in 46.7% of respondents aged over 65 years. There has also been observed a correlation between physical activity and the prevalence of the seven most common non-communicable diseases, often associated with aging.

Depression remains a serious health problem for older people, leading to higher risk of cognitive decline and lower quality of life [27]. The COVID-19 pandemic and subsequent isolation have been shown to further impact the psychological well-being of older people [28]. In this study, subjects from the most active group rated their psychological well-being significantly higher than the ones who were the least physically active. Also, the prevalence of depression was the lowest in older people who engaged in high level of physical activity. Studies show that regular aerobic exercise lowers the risk for developing depression among older individuals [29,30].

Although physical activity is beneficial for people of all ages, older people tend to become less active as they progress in age [31]. The ongoing pandemic is another factor that plays into the diminishing number of active seniors [32], not only impeding them from participating in exercise classes or using the gym, but also community activities such as shopping and socializing [33]. Visser et al. found that, during the COVID-19 pandemic, 36.4% of people over the age of 55 years (mean age = 74, sample size *n* = 1027) met the recommendation of 150 min of physical activity (of at least moderate-intensity) a week and about 50% of the subjects experienced a decline in physical activity and exercise during the pandemic [34]. Lesser and Nienhuis found that among older Canadians, 36.6% were physically active during the pandemic, meaning they met the same recommendations as previous study [35]. Japanese adults over 65 years old who participated in an online survey had a median time of total physical activity per week of 180 min (added light, moderate and vigorous activity) in April 2020, which was lower than pre-pandemic January 2020: 245 min [36]. Agrawal et al. found that over 55% of 500 older Polish people limited their recreational activities due to the fear of infection with SARS-CoV-2 [37]. This is especially worrying given the fact that the population of older people in Poland has already had a low level of physical activity [38]. Washif et al. found that even among younger people, the pandemic has negatively impacted their physical activity, reducing frequency, duration and intensity of training. This was observed even among world-class and international athletes [39]. Similar findings have been presented by Trabelsi et al.: during the COVID-19 home confinement, time spent engaging in physical activity decreased significantly and number of hours spent sitting increased by ~2 h/day [40]. Additionally, returning to physical activity after COVID-19 illness should be done in a careful manner as it can potentially lead to cardiac injury or thromboembolic events [41]. Janovský et al. analyzed changes in physical activity of 204 older people (average age 84.55 years) from Czech Republic between the pre-pandemic and during-pandemic time. According to their tracker-based measurement, the majority of seniors experienced a decline in their physical activity. However, a group of subjects (42%) experienced an increase in their physical activity during the pandemic when compared to the same month in the year prior [42]. A meta-analysis of 173 studies revealed that, during COVID-19 lockdowns, physical activity decreased and sedentary behavior increased at the same time [43]. Another study from South Africa showed that moderate to high levels of physical activity may have significant benefits especially in preventing negative outcomes of COVID-19, such as hospitalization, ICU admission, ventilation and death [44]. We found that 21.9% of older people engaged in at least 2 h (120 min) of physical activity per week. GUS reports that although 25.1% of people over 60 years old in Poland engaged in physical activity and exercise activities, only 10.6% declared doing so regularly/often [45]. Being physically active on most days of week was declared by 36% of people aged 65–74 years in the WOBASZ study [46]. Due to different research approaches and varying sample sizes, there are differences in findings about the level of physical activity in older Polish people.

### Strengths and Limitations

This study is based on the data collected during the National Test for Poles’ Health , which is an online survey. One of the advantages of this method of collecting information is the accessibility for potential participants, especially because it has been published on one of the biggest health-oriented Internet platforms in Poland, Medonet. This ensures a large sample size, but it also comes with some disadvantages, such as the inability to verify the credibility of respondent’s identity and their answers. Though the questions have been formulated to be easily understood for everyone, there is always a possibility some respondents do not fully comprehend them, impacting the way they answer. At the time of conducting this study, online survey was the safest method of gathering information from a large group of respondents in the time of necessary pandemic-related restrictions and social-distancing measures. The authors also acknowledge the fact that the validity of the questionnaire was not measured.

The NTZP has by far gathered information on health and physical activity from the largest older people population, not only in Poland or the Central-Eastern Europe region, but possibly on the world scale. Therefore, the unquestionable strength of this study is the sample size: 64,732 older people (65+ years old), making it the biggest study of health and well-being of people in Poland during the COVID-19 pandemic. However, the authors acknowledge that this population of older people include only ones that use the Internet, as this was an online type of study. Another disadvantage is that the questionnaires were completed by respondents without supervision and all answers (including weight and height) were not possible to be verified. The intensity of physical activity was also not measured by the questionnaire. This analysis focuses on physical activity defined as time spent engaging in sport and exercise. Walking was not included in this analysis, which may have impacted the results. After completing the questionnaire, respondents were given an estimated biological age. As mentioned before, this estimation was based on a calculator by Thomas Perls [17]. However, this tool has not been tested with validity and repeatability studies and only serves as a guide and encourages respondents to improve their lifestyle. Therefore, it has not been included in this analysis. These aspects should be taken into consideration by the readers.

Based on the cross-sectional type of our study, we cannot conclude about the cause/effect of regular physical activity on the physical and psychological health of older people. Based on our own longitudinal prospective studies in middle-aged men with more than 15 years observation, diverse positive health effects of regular physical activity in older population seem very possible [47,48,49]. Further longitudinal, prospective studies are necessary in order to demonstrate a causal relationship between regular physical activity in older men and women and beneficial health effects.

## 5. Conclusions

This study provides information on the health behaviors of Polish seniors during the COVID-19 pandemic, which has had an especially significant impact on this country. It is one of the biggest studies on this matter conducted during the pandemic, which makes it a valuable source of novel information.

This study showed that older people who engaged in at least 1 h of physical activity a week had statistically significant better health status (measured by subjective evaluation of both physical and psychological health, as well as the prevalence of multimorbidity) than those who did not. Subjects from the most active group (engaging in at least 2 h of physical activity/week) have shown to be in better health than the other two groups. Findings from this study suggest that the majority of older Internet users in Poland do not meet the recommended level of physical activity for older people [33]. These outcomes may serve as an important tool for improving well-being in this population during the global pandemic. Therefore, it should be the priority for public health policymakers to focus on ways to encourage seniors to participate in physical activity, especially the kinds that are safe during the COVID-19 pandemic.

## Figures and Tables

**Figure 1 ijerph-20-00556-f001:**
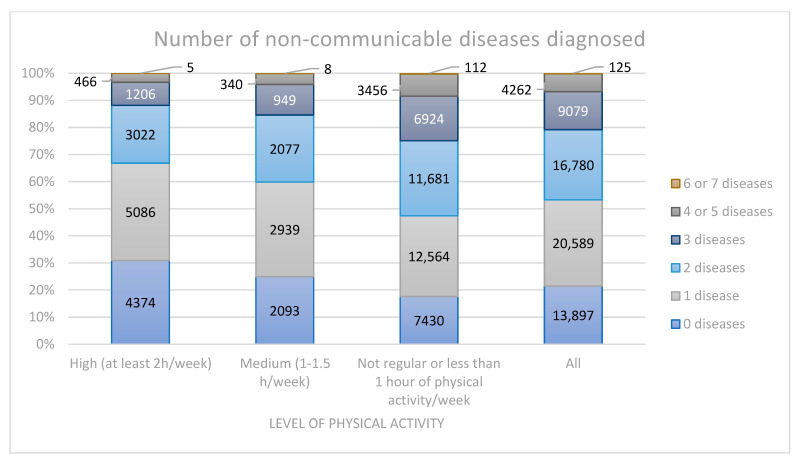
Number of non-communicable diseases diagnosed.

**Figure 2 ijerph-20-00556-f002:**
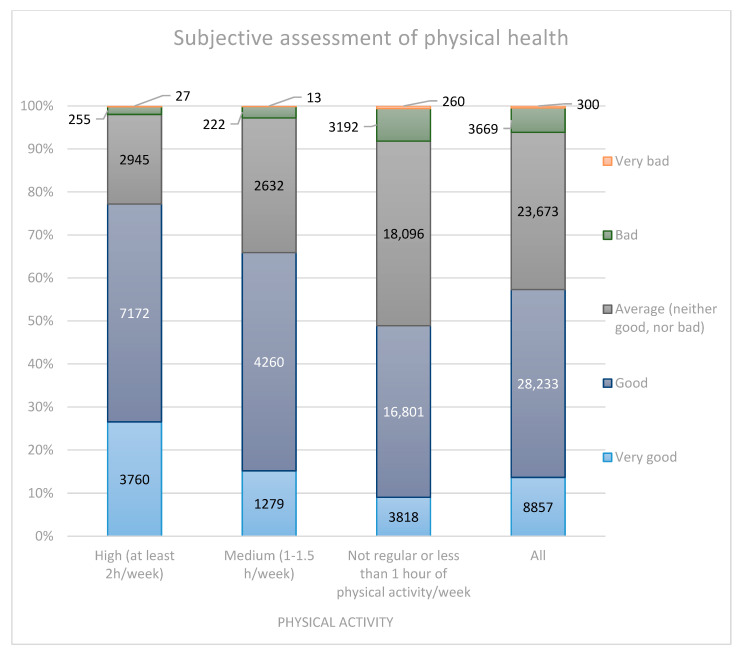
Subjective assessment of physical health.

**Figure 3 ijerph-20-00556-f003:**
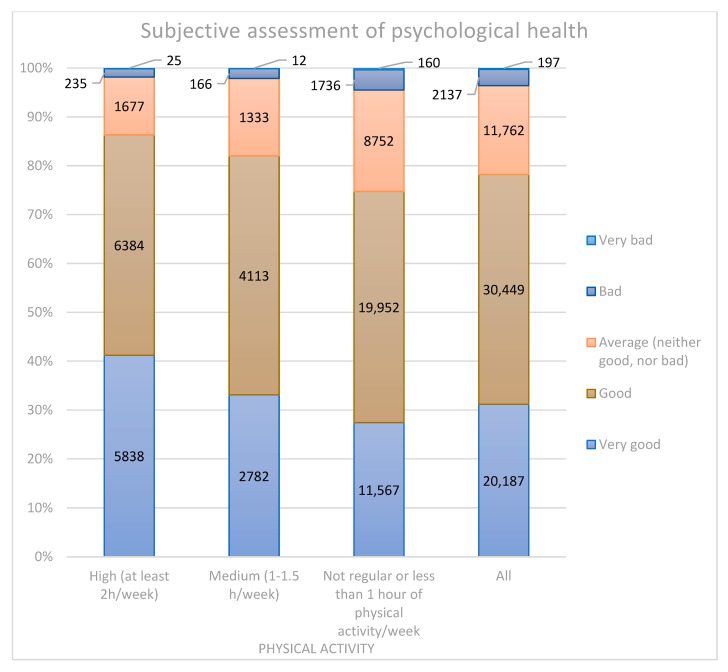
Subjective assessment of psychological health.

**Table 1 ijerph-20-00556-t001:** Characteristics of the subjects.

	*n*	Percentage
Gender:		
Female	31,352	48.4%
Male	33,380	51.6%
Age:		
65–69	34,473	53.3%
70–74	20,696	32.0%
75–79	6500	10.0%
80–84	2340	3.6%
85+	723	1.1%
Place of residence:		
Village	10,371	16.0%
Town, less than 19,000 inhabitants	7860	12.1%
Town, between 20,000 to 49,000 inhabitants	10,593	16.4%
Town, between 50,000 to 99,000 inhabitants	8545	13.2%
Town, between 100,000 to 199,000 inhabitants	8545	13.2%
Town, between 200,000 to 499,000 inhabitants	8271	12.8%
Town, more than 500,000 inhabitants	10,547	16.3%
Education:		
Primary	868	1.3%
Junior-high	261	0.4%
Vocational	5483	8.5%
Secondary	23,042	35.6%
Post-secondary	8080	12.5%
Bachelor’s degree/engineer	5217	8.1%
Master’s degree	21,781	33.6%

**Table 2 ijerph-20-00556-t002:** Respondents aged 65+ years old according to sex and their level of physical activity.

Physical Activity	Sex					
	Female		Male		All	
	*n*	% 95% CI	*n*	% 95% CI	*n*	% 95% CI
High (at least 2 h of physical activity/week)	5663	18.1% *** (17.7–18.5)	8496	25.5% (25.0–26.0)	14,159	21.9% (21.6–22.2)
Medium (1–1.5 h of physical activity/week)	3982	12.7% * (12.3–13.1)	4424	13.3% (12.9–13.7)	8406	13.0% (12.7–13.3)
Not regular or less than 1 h of physical activity/week	21,707	69.2% *** (68.7–69.7)	20,460	61.2% (60.7–61.7)	42,167	65.1% (64.7–65.5)
All	31,352	100.0%	33,380	100.0%	64,732	100.0%

* *p* < 0.05; *** *p* < 0.001 Female vs. Male.

**Table 3 ijerph-20-00556-t003:** Prevalence of physical activity on the form of active sport participation and walking.

Time Spent Walking			Physical Activity					
	High (At Least 2 h of Physical Activity/Week)		Medium (1–1.5 h of Physical Activity/Week)		Not Regular or Less Than 1 h of Physical Activity/Week		All	
	*n*	%	*n*	%	*n*	%	%	%
The respondent does not leave the house	276	1.9	225	2.7	5016	11.9	5517	8.5
Under 30 min/day	1740	12.3	1713	20.4	11,891	28.2	15,344	23.7
30 to 60 min/day	4854	34.3	3404	40.5	13,873	32.9	22,131	34.2
1 to 2 h/day	4880	34.5	2273	27.0	8233	19.5	15,386	23.8
More than 2 h/day	2409	17.0	791	9.4	3154	7.5	6354	9.8
All	14,159	100.0	8406	100.0	42,167	100.0	64,732	100.0

**Table 4 ijerph-20-00556-t004:** Physical activity and BMI (all).

Physical Activity	*n*	Mean BMI in kg/m^2^	Confidence −95.000%	Confidence 95.000%	Minimum	Maximum	Standard Deviation
high (at least 2 h/week)	14,159	27.0	26.9	27.0	14.2	58.8	3.8
medium (1–1.5 h/week)	8406	27.5	27.4	27.6	16.5	59.5	4.1
Not regular or less than 1 h of physical activity/week	42,167	28.4	28.4	28.5	13.6	59.9	4.6
All	64,732	28.0	28.0	28.0	13.61	59.9	4.4

ANOVA F = 657.059, *p* < 0.001. All differences between mean BMI (for specific physical activity level groups) are statistically significant with *p* < 0.001.

**Table 5 ijerph-20-00556-t005:** Physical activity and BMI (women).

Physical Activity	*n*	Mean BMI in kg/m^2^	Confidence −95.000%	Confidence 95.000%	Minimum	Maximum	Standard Deviation
high (at least 2 h/week)	5663	26.5	26.4	26.6	14.2	58.8	4.1
medium (1–1.5 h/week)	3982	27.0	26.9	27.2	16.7	59.4	4.3
Not regular or less than 1 h of physical activity/week	21,707	28.2	28.1	28.2	13.6	59.9	4.9
all	31,352	27.7	27.7	27.8	13.6	59.9	4.7

ANOVA F = 333.036, *p* < 0.001. All differences between mean BMI (for specific physical activity level groups) are statistically significant with *p* < 0.001.

**Table 6 ijerph-20-00556-t006:** Physical activity and BMI (men).

Physical Activity	*n*	Mean BMI in kg/m^2^	Confidence −95.000%	Confidence 95.000%	Minimum	Maximum	Standard Deviation
High (at least 2 h of physical activity/week)	8496	27.3	27.2	27.4	14.7	57.5	3.5
Medium (1–1.5 h of physical activity/week)	4424	27.9	27.8	28.0	16.5	59.5	3.8
Not regular or	20,460	28.7	28.7	28.8	13.6	59.8	4.3
less than 1 h of physical activity/week							
All	33,380	28.3	28.2	28.3	13.6	59.8	4.1

ANOVA F = 397.862, *p* < 0.001. All differences between mean BMI (for specific physical activity level groups) are statistically significant with *p* < 0.001.

**Table 7 ijerph-20-00556-t007:** Prevalence of diseases in different physical activity groups.

Disease	Physical Activity						All	
	High (At Least 2 h/Week)		Medium (1–1.5 h/Week)		Not Regular or Less Than 1 h of Physical Activity/Week			
	*n*	% 95% CI	*n*	% 95% CI	*n*	% 95% CI	*n*	% 95% CI
Obesity	2579	18.2% *** ^###^ (17.6–18.9)	1953	23.2% ^&&&^ (22.3–24.1)	13,667	32.4% (32.0–32.9)	18,199	28.1% (27.8–28.5)
Hypertension	6943	49.0% *** ^###^ (48.2–49.9)	4562	54.3% ^&&&^ (53.2–55.3)	25,428	60.3% (59.8–60.8)	36,933	57.1% (56.7–57.4)
Diabetes	1932	13.6% *** ^###^ (13.1–14.2)	1311	15.6% ^&&&^ (14.8–16.4)	9301	22.1% (21.7–22.5)	12,544	19.4% (19.1–19.7)
Heart diseases	2826	20.0% *** ^###^ (19.3–20.6)	1943	23.1% ^&&&^ (22.2–24.0)	11,976	28.4% (28.0–28.8)	16,745	25.9% (25.5–26.2)
COPD	526	3.7% ** ^###^ (3.4–4.0)	379	4.5% ^&&&^ (4.1–5.0)	3187	7.6% (7.3–7.8)	4092	6.3% (6.1–6.5)
Depression	708	5.0% *** ^###^ (4.6–5.4)	502	6.0% ^&&&^ (5.5–6.5)	3807	9.0% (8.8–9.3)	5017	7.8% (7.5–8.0)
Cancer	1230	8.7% ^###^ (8.2–9.2)	765	9.1% ^&&&^ (8.5–9.7)	4493	10.7% (10.4–10.9)	6488	10.0% (9.8–10.3)

** *p* < 0.01; *** *p* < 0.001 high physical activity vs. medium physical activity; ^###^
*p* < 0.001 high physical activity vs. none; ^&&&^
*p* < 0.001 medium physical activity vs. none.

**Table 8 ijerph-20-00556-t008:** Prevalence of multimorbidity.

	Sex	Physical Activity						All	
		High (At Least 2 h/Week)		Medium (1–1.5 h/Week)		Not Regular or Less Than 1 h of Physical Activity/Week			
		*n*	% 95% CI	*n*	% 95% CI	*n*	% 95% CI	*n*	% 95% CI
Multimorbidity presence (2 or more non-communicable diseases)	Female	1723	30.4% *** ^###^ (29.2–31.6)	1432	36.0% ^&&&^ (34.5–37.5)	10,776	49.6% (49.0–50.3)	13,931	44.4% (43.4–44.5)
	Male	2976	35.0% *** ^###^ (34.0–36.0)	1942	43.9% ^&&&^ (42.4–45.4)	11,397	55.7% (55.0–56.4)	16,315	48.9% (48.3–49.4)
	All	4699	33.2% *** ^###^ (32.4–34.0)	3374	40.1% ^&&&^ (39.1–41.2)	22,173	52.6% (52.1–53.1)	30,246	46.7% (46.3–47.1)
	*p* (Female vs. Male)	*p* < 0.001		*p* < 0.001		*p* < 0.001		*p* < 0.001	

*** *p* < 0.001 high physical activity vs. medium physical activity; ^###^
*p* < 0.001 high physical activity vs. none; ^&&&^
*p* < 0.001 medium physical activity vs. none.

## Data Availability

Data supporting reported results can be found at https://narodowytestzdrowia.medonet.pl/ (accessed on 26 December 2022).

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
