# Peer review of "Physical Activity versus Selected Health Behaviors, Subjective Physical and Psychological Health and Multimorbidity in a Large Cohort of Polish Seniors during the COVID-19 Pandemic (Results of the National Test for Poles’ Health )"

_ijerph, 2022, doi:10.3390/ijerph20010556_

Round 1

Reviewer 1 Report

The work presented to me for the review is an interesting study showing various aspects of physical activity of Polish seniors during the COVID-19 pandemic. The manuscript is well written and, in my opinion, methodically correct. Introduction is short and concise. Results are presented in a clear way, Discussion and Conclusions are adequate. Strenghts and limitations section is an important part of the manuscript - it's very aptly written.

Please find my minor comments below:

Lines 36-38: keywords are missing

Lines 64-66: please explain the term "strengthen the immune system" - it's too general

Methods: Please add the information in which form data from NTZP is available - is the access public or on demand?

Line 86: "both waves"? please specify - did you mean both waves of COVID-19 pandemic or you meant 2 editions of NTZP?

Line 120: please add information about version and manufacturer of Statistica software

Lines 123-125: all the differences in physical activity level between men and women shown in the Table 1 came out statistically significant, and you describe only the first part - I suggest to provide p values for all data from this table.

Author Response

Dear reviewer,

Thank you for your remarks and comments, which undeniably contributed to improving this work’s quality.

We have added the keywords and explained the term "strengthen the immune system, adding references as well.

We have also added the information about the second wave of NTZP and the manufacturer of Statistica software, which is Dell Inc.

We have also changed the decription of statistical methods used.

Thank you again for your comments.

Sincerely,

Agnieszka Szychowska

Reviewer 2 Report

The paper aims to outline the outcomes of a large online survey conducted in a population of polish seniors and draw conclusions about the benefits of physical activity on health-related parameters, morbidity and self-perceived status of health during the covid-19 pandemic.

The main findings of the study show that an engagement in at least 2 hours of physical activity/week is linked to a better perceived physical and psychological health and lower prevalence of multimorbidity in people aged 65 and above.

These outcomes may serve as an important tool for improving well-being in this population during the global pandemic. However, the health benefits of a physically active lifestyle cannot be seen as a novel finding and there were some major methodological flaws identified.

The National Polish Health Test is briefly described in the introduction, only little additional information about the survey is provided in the section “Material and Methods” and there was no supplementary material accessible. Thus the methodology, specifically regarding the questionnaire that was used, is described insufficiently.

For example, the differentiation of physical activity levels is described in lines 97-100. It is not adequately specified what was being defined as “physical activity” (PA) and how respondents were instructed to rate their answers.

There is no information about the intensity or type of PA provided.

Furthermore, it is unclear how and why the three PA subgroups for the statistical analysis were chosen. It is pointed out that people who do not exercise at all and people who engage in less than 1 hour of physical activity a week were combined to one group because these two subgroups showed very similar results. This decision has seemingly been made after the data analysis which carries certain risks for bias and should be interpreted with caution.

Similarly to the vague description of the assessment of PA, there was no clear definition given of the rating of psychological and physical health. Participants rated on a scale from „very good” to „very bad” (lines 106-107) with no indication of how many answers in between these extremes were possible.

The statistical analysis is not appropriate for such a large-scale epidemiological study. It is not sufficiently explained how and why the specific methods have been chosen.

To point out an example, it seems highly unlikely that the Mann-Withney test, as a nonparametric test, serves as the best option to draw comparisons between two groups in such a large cohort. Moreover, it was not specified which post-hoc or multiple comparison test was used after the ANOVA.

In lines 118-121 the p-values for statistically significant results are indicated. While it is first stated that The differences were considered statistically significant for p <0.05, it is shortly after described as A p-value of <0.001 or <0,05 was considered to be statistically significant, depending on the characteristic, which is ambiguous and needs further explanation.

The results are partly displayed in an unclear and not well formatted way, for example in tables 3-5 (in each of the first lines words are cut off) and tables 6 and 7 (too many group comparisons in one table and confusing labeling of p-values).

Statistically significant results were not always highlighted in the tables, for example in table 3, where the indication of significant differences in BMI between PA-groups cannot be seen (but were pointed out before in the text). The visual presentation of the outcomes regarding the number of non-communicable diseases and the subjective assessments of health are not formatted adequately.

The points mentioned above were not touched upon in the discussion or mentioned as limitations of the study.

Furthermore, the results cannot be highlighted as new findings and the authors should clearly point our which was the novelty of this study compared to previous ivestigations .

Author Response

Dear reviewer,

Thank you for the valuable reviews which undeniably contributed to improving this work’s quality. We have applied the proposed changes to the manuscript.

In the statistical analysis non-parametric significance test were used due the fact that the distribution of the analyzed features differed from the normal distribution.

Taking into account the fact that for large groups of several hundred units, it can be concluded that the distribution of features is close to the normal distribution, the significance of differences in average values was recalculated; to compare the significance of differences in mean values in two groups, the Student's t-test for independent samples was used, and in three groups, the analysis of variance ANOVA with the Sheffe post-hoc test was used. The results obtained as a result of the application of parametric significance tests confirmed (were consistent) with the results obtained as a result of the analyzes carried out using non-parametric significance tests.

We agree that online surveys come with methodological limitations, however they have been mentioned and described in the methods description, as well as in the Strengths and limitations part. This was the safest method of gathering information from a large group of respondents in the time of necessary pandemic-related restrictions and social-distancing measures.

We have explained the rating of psychological and physical health, which included five possible answers for the repsondents to choose from. We also explained further the question about physical activity.

This study provides information on the health behaviors of older Internet users in Poland during the COVID-19 pandemic, which has had an especially significant negative health impact on this country. It is one of the biggest studies on this matter conducted during the pandemic, which makes it a valuable source of novel information.

Thank you again for your comments.

Reviewer 3 Report

The title does not support the findings of this research.

Subjective physical health parameters do not elaborate on in this article.

Subjective Physiological health measurements do not mention in this article. 

No details were found regarding the questionnaire.

The reliability and validity of the questionnaire OR data were not measured.

To determine psychological health: what parameters were used? 

BMI was calculated based on the respondents’ answers about their weight and height. Validity and reliability? 

Online surveys have poor response rates leading to bias affecting the validity and generalizability.

The Mann-Whitney test was used to assess the differences between the mean values in the two groups. (is that True? )

The Mann-Whitney test compares the mean ranks.

Author Response

Dear reviewer,

Thank you for your remarks and comments, which undeniably contributed to improving this work’s quality.

Our impression is that the title describes the topic of this manuscript well, thus we decided to not apply any changes to it.

We agree that online surveys come with methodological limitations, however they have been mentioned and described in the methods description, as well as in the Strengths and limitations part. This was the safest method of gathering information from a large group of respondents in the time of necessary pandemic-related restrictions and social-distancing measures.

We have also changed the decription of statistical methods used.

Thank you again for your comments.

Sincerely,

Agnieszka Szychowska

Reviewer 4 Report

Thank you for the opportunity to review this article. The paper addresses a novel under-researched area, which has the potential to provide useful recommendations for coaches. However, there are some questions that need to be addressed to the manuscript.

Specific comments are provided below:

TITLE

The title is too long. Could you summarize it? 

ABSTRACT

Name “COPD” (line 25) 

Name “BMI” (line 26)

All decimal numbers with ".". Review the entire document.

Include keywords (line 36)

INTRODUCTION

“Therefore” before “it is” (line 43)

Name “US” (line 51)

Name “CVD” (line 53)

This sentence is not relevant. Please, include the website as reference. (line 77-79)

MATERIAL AND METHODS

Explain before the year of each wave (line 86)

Please, not repeat the information of table 1. Include only this information in table 1. (line 92-93)

Try to write more easily this sentence (line 93)

Name “COPD” (line 104)

Include a statistical analysis section (line 108)

Could you combine table 3,4, and 5? There are many tables and it is difficult to read the content.

Name “COPD” in the legend table 6. 

p value always in italics. Review the entire document.  

Quotation marks at top. (line 167). Review the entire document.  

Add figure 1, 2 and 3 in the text. 

Can you add p value in the figures? 

DISCUSSION

Add references (line 184-185)

Add “,” after “however”. (line 189)

Further analyze these results in relation to the scientific literature. (line 206-208)

Only includes one reference.... please include more references to say "studies". (line 216-217)

Remove “:” (line 238)

Explain further why you obtained your results and why they are similar or different from those of other studies. (line 218-253)

Author Response

Dear reviewer,

Thank you for the valuable reviews which undeniably contributed to improving this work’s quality. 

Our impression is that the title describes the topic of this manuscript well, thus we decided to not apply any changes to it.

We have explained the abbreviations and applied the punctuation changes.

We have also changed the decription of statistical methods used and added references where needed. We added a brief decription of two other studies, which decribed the effect that the pandemic had on the level of physical activity in people all ages.

Thank you again for your comments.

Reviewer 5 Report

Line 96-98, how did the authors define the level of physical activity? and please give the references.

Also line 106; how did the authors provide or classify the subjective physical and psychological health? 

Line 129, please provide the unit of BMI and also other measurements. 

Line 193; please clarified the obesity paradox.

Author Response

Dear reviewer,

Thank you for the valuable reviews which undeniably contributed to improving this work’s quality. 

We agree that online surveys come with methodological limitations, however they have been mentioned and described in the methods description, as well as in the Strengths and limitations part. This was the safest method of gathering information from a large group of respondents in the time of necessary pandemic-related restrictions and social-distancing measures.

We have added the information on assessing the subjective physical and psychological health by the respondents, as well as how physical activity was defined.

The obesity paradox is decribed in the paragraph and suitable references are mentioned.

Thank you again for your comments.

Round 2

Reviewer 2 Report

The clearification of the methodological flaws that were pointed out certainly helped to improve the quality of the work and make readers aware of its limitations.

In line with this transperancy it should also be mentioned in the "Strenght and limitations" chapter that not including walking as a form of PA in the analysis could have impacted the results.

It would also be important to clearify why walking was not treated as a form of PA in this analysis given it's beneficial effect on health-related parameters that have been demonstrated in multiple studies. 

The display of results has been worked on, however the quality of the visual presentation of results (especially the results of non-communicable diseases, physical and psychological health) should to be further improved before publication. 

Author Response

Dear Reviewer,

We would like to thank again for the valuable reviews. We have applied the proposed changes to the manuscript. We have worked on the  visual presentation of results, hoping they are now more clear for the reader.

This analysis focuses on time spent engaging in sport and exercising by the older people. It does not take into consideration time spent walking or other types of physical activity, which is a subject of another article, that the authors hope to publish in the near future. 

Thank you again for your review.

Reviewer 3 Report

The title does not support the findings of this research. Because…

(The title of the manuscript is "Physical activity versus selected health behaviours, subjective physical and physiological health and multimorbidity in a  large cohort of Polish Seniors during COVID-19 pandemics" )

It contains:

1. Physical activity: high level of physical activity (at least 2 hours/week), medium (1-1,5 hours/week).

2. selected health behaviours: non-communicable diseases: obesity, hypertension, diabetes, heart diseases, chronic obstructive pulmonary disease (COPD), depression, and cancer.

3. subjective physical: What are the parameters used for analysis?

4. physiological health:  What are the parameters used to analysis?

5. and multimorbidity: What are the parameters used for analysis?

The title does not mention psychological health but its included in the analysis.

No information was provided on all these parameters in the methodology.

How the Subjective assessment of physical health determine

How the Subjective assessment of psychological health determine

Subjective physical health parameters do not elaborate on in this article.

Subjective Physiological health measurements do not mention in this article. 

No details were found regarding the questionnaire.

Kindly submit a copy of the questionnaires as supplementary material.

The reliability and validity of the questionnaire OR data were not measured.

To determine psychological health: what parameters were used? 

BMI was calculated based on the respondents’ answers about their weight and height.

Why BMI categories were not determined, such as underweight, Healthy, Overweight, and obese with class I, II, & III? 

The minimum score for BMI is 13.6 which represents underweight; the Mean value is 26.5, which represents most of the participants were overweight, and the maximum score is 59.9, which means obese with level 3 at high risk.  

Classifications of BMI level with physical activity level, Prevalence of diseases, and Prevalence of multimorbidity may be worthwhile for the analysis.  

The Mann-Whitney test was used to assess the differences between the mean values in the two groups. (is that True? )

The Mann-Whitney test compares the mean ranks.

The statistical analysis and results do not show consistency

As table# 3 showed Min.=14.2, Max.=58.2 Mean=27.0 Confidence 95% =26.9 and 27, than,
How standard deviation is 3.8 for BMI at a high level of physical activity?

All the standard deviation values follow the same pattern, which is not feasible.

 As mentioned in the statistical analysis, Student's t-test for independent samples was used, and in three groups, the analysis of variance (ANOVA) with the Sheffie posthoc test was used

t-value, and p-value should be provided.
If the ANOVA result is significant, then the Sheffe post-hoc test table should be provided. Effect size is also calculated.

Which test was applied for tables #6 and 7? Mention it, How there is significant ** p<0,01; *** p<0,001 high physical activity vs medium physical activity

In Tables 6 & 7, 95% Class Interval (CI) values represent what? It needs a different analysis test if it is a discrete variable.  

Kindly submit a copy of the SPSS sheet with data as supplementary material.

Author Response

Dear Reviewer,

We would like to thank again for the valuable reviews. 

We corrected the title according to your remarks. We also added supplementary materials which should provide further information about this study’s methodology for the readers. We believe adding these changes substantially improves the article’s quality.

This analysis focuses on time spent engaging in sport and exercising by the older people. It does not take into consideration time spent walking or other types of physical activity, which is a subject of another article, that the authors hope to publish in the near future. Classification of BMI levels will also be taken into consideration.

We have controlled all statistical results and haven’t found any errors. The test for two fractions from large samples was used to assess the significance of differences between the frequencies of the categories of non-measurable variables in two groups. Effect size was also calculated, as requested by the reviewer. Following the remarks, we submit a copy of the SPSS sheet with data.

Thank you again for the valuable review.